# Vascular Damage and Glycometabolic Control in Older Patients with Type 2 Diabetes

**DOI:** 10.3390/metabo13030382

**Published:** 2023-03-03

**Authors:** David Karasek, Jaromira Spurna, Dominika Macakova, Ondrej Krystynik, Veronika Kucerova

**Affiliations:** 13rd Department of Internal Medicine—Nephrology, Rheumatology and Endocrinology, University Hospital Olomouc, Faculty of Medicine, Dentistry Palacky University Olomouc, 77900 Olomouc, Czech Republic; 2Department of Clinical Biochemistry, University Hospital Olomouc, 77900 Olomouc, Czech Republic

**Keywords:** type 2 diabetes, elderly, arterial stiffness, von Willebrand factor, tissue plasminogen activator, plasminogen activator inhibitor-1

## Abstract

Diabetes is one of the main risk factors for vascular damage, including endothelial dysfunction and arterial stiffness. The aim of this study was to compare selected parameters of vascular damage in patients with type 2 diabetes (T2D) in different age categories and to determine their relationship to indicators of glycometabolic control. A total of 160 patients with T2D were included in this cross-sectional study. They were divided into four age quartiles (with mean ages of 42.1 ± 4.5, 51.6 ± 1.4, 59.2 ± 3.0, and 69.8 ± 3.8, respectively). All subjects were evaluated for indicators of glycometabolic control and for arterial stiffness parameters along with markers of endothelial damage—tissue plasminogen activator (tPA), plasminogen activator inhibitor-1 (PAI-1) and von Willebrand factor (vWF). The oldest compared to the youngest participants showed significantly increased parameters of arterial stiffness (augmentation pressure 13.4 ± 8.6 vs. 6.7 ± 4.4 mm Hg, augmentation index 26.2 ± 11.3 vs. 19.6 ± 9.2 mm Hg, aortic pulse pressure 47.7 ± 17.1 vs. 33.7 ± 10.4 mm Hg, and pulse wave velocity 11.9 (10.1–14.3) vs. 8.2 (7.7–9.8) m/s) despite having similar glycometabolic control. Arterial stiffness parameters were mainly associated with age and blood pressure. Age and systolic blood pressure were major determinants of arterial stiffness regardless of glycometabolic control. The oldest patients also had the highest levels of vWF (153.7 ± 51.9 vs. 121.7 ± 42.5 %) but the lowest levels of PAI-1 (81.8 ± 47.5 vs. 90.0 ± 44.9 ng/mL). Markers of endothelial dysfunction correlated with metabolic parameters, but did not correlate with arterial stiffness. Age and systolic blood pressure are major determinants of arterial stiffness in patients with T2D regardless of glycometabolic control, whereas an unfavorable metabolic profile is mainly related to endothelial dysfunction. These results suggest a differential contribution of cardiometabolic risk factors to vascular damage in T2D patients over their lifetime.

## 1. Introduction

In recent decades, the incidence of diabetes has been increasing worldwide, especially among the elderly. An aging population is considered one of the most significant contributors to the increasing prevalence of diabetes. In the US, more than one-third of the adult population with diabetes is currently 65 years of age and older [1]. In high-income countries, the prevalence of diabetes peaks (22%) in the 75–79 age group and in middle-income countries in the 60–74 age group (19%) [2]. Diabetes and the aging process independently increase the risk of cardiovascular disease (CVD). Elderly diabetic patients show higher vascular damage and CVD risk than those without diabetes. Vascular inflammation and oxidative stress appear to play a major role in the mechanisms of aging, diabetes and CVD [3]. However, the precise mechanisms underlying age- and diabetes- related CVD remain poorly understood, including the contribution of glycometabolic control.

Vascular stiffness represents a subclinical marker of CVD risk. Both age and diabetes are important determinants of vascular damage [4,5,6]. Several studies have found that arterial stiffness may be a predictor of future CVD morbidity and mortality in the diabetic population [7,8]. It is important to note that an independent relationship between arterial stiffness and diabetes has not been consistently demonstrated in all studies. Diabetes does not appear to be a major determinant for this type of vascular damage, especially in older, hypertensive patients [7,9]. On the other hand, some findings suggest that vascular stiffness in diabetic patients may be attributed to the role of diabetes itself rather than aging and higher blood pressure [4]. These possible metabolic mechanisms include non-enzymatic advanced glycation of proteins with production of advanced glycation end products leading to an abnormal extracellular matrix, endothelial dysfunction with nitric oxide dysregulation associated with a tendency to vasospasm, and/or chronic vascular inflammation accompanied by accelerated arterial calcification [7,10].

Both arterial stiffness and endothelial dysfunction represent surrogate markers for CVD; however, they reflect different aspects of vascular damage [11]. Arterial stiffness results mainly from arteriosclerosis (primary disease of the media); endothelial dysfunction contributes to atherosclerosis (primary disease of the intima). Traditional CVD risk factors, such as hypertension, dyslipidemia or diabetes, may differentially affect vascular involvement. Some drugs, such as statins and angiotensin-converting enzyme (ACE) inhibitors, can simultaneously improve endothelial function and reduce arterial stiffness. Glucagon-like peptide 1 (GLP-1) receptor agonists and gliflozins also have a positive effect on arterial stiffness and restore endothelial function [12]. Recently, GLP-1 receptor agonists, gliflozin, and especially their combination have shown greater reductions in markers of endothelial dysfunction and arterial stiffness than insulin in patients with T2D, despite similar reductions in glycosylated hemoglobin [13]. Thus, the contribution of glycometabolic control to vascular damage in T2D is not always the same and may change throughout life.

The aim of this study was to compare markers of glycometabolic control and vascular damage in patients with type 2 diabetes (T2D) according to age and to determine whether the parameters of arterial stiffness and/or endothelial dysfunction are influenced by the glycometabolic control.

## 2. Materials and Methods

### 2.1. Study Design and Subjects

This cross-sectional study comprised T2D patients consecutively examined during their visits in an outpatient diabetic clinic. The principles of the Declaration of Helsinki for human experiments were respected. The study design and informed consent were reviewed and approved by the Ethics Committee of the Faculty of Medicine and University Hospital Olomouc. All participants were asked about their previous medical history, especially their cardiovascular status, medication, diabetic complications and diabetes duration. We used the following criteria to diagnose diabetes: fasting plasma glucose level ≥7 mmol/L and/or oral antidiabetic drugs (OADs) or insulin administration. Subjects with type 1 diabetes, secondary or genetic diabetes, infection, active cancer, and trauma were not included in this study. Body mass index (BMI), waist circumference, systolic and diastolic blood pressure (SBP and DBP) were also measured. BMI was calculated as body weight/body height^2^ (kg/m^2^). Waist circumference was measured while standing, in the middle between the anterior iliac crest and the lower border of the ribs.

### 2.2. Arterial Stiffness Measurements

These markers of arterial stiffness were used: augmentation index (AIx), augmentation index normalized for a heart rate of 75 beats per minute (AIx-75), augmentation pressure (AP), aortic systolic pressure (Aortic SP), aortic pulse pressure (Aortic PP), and pulse wave velocity (PWV) [14,15]. The measurement was performed with the SphygmoCor system (AtCor Medical Pty Ltd. Head Office, West Ryde, Sydney, Australia). At least 12 h before the examination, patients were not allowed to smoke or drink alcohol or caffeinated beverages. They were examined in the morning after at least 10 min of rest in a quiet, temperature-controlled room. The examination first took place in a sitting position with a sensor on the radial artery to estimate the aortic pulse wave. PWV was then measured in the supine position; carotid and femoral artery pulse waves were analyzed, and the delay with respect to the ECG wave was detected. Integral software was used to process each pulse wave and ECG data set to analyze the average time difference between the R-wave and the pulse wave over about 10 consecutive cardiac cycles.

Distance measurements were made using a tape measure from the sternum (carotid site) to the femoral arteries at the sensor site. Subsequently, PWV was calculated using the distance and average time difference between the two recorded sites according to the formula: PWV (m/s) = carotid–femoral distance (m)/carotid–femoral transit time (s).

### 2.3. Laboratory Analyses

Venous blood samples were drawn in the morning after a 12 h fast. Routine serum biochemical parameters were analyzed on the day of blood collection. The modular system SWA (Serum Work Area, Roche, Basel, Switzerland) was used for biochemical examinations. Total cholesterol (TC), TG, and high-density lipoprotein cholesterol (HDL-C) were determined enzymatically. Low-density lipoprotein cholesterol (LDL-C) was calculated according to the Friedewald formula (LDL-C = TC—TG*0.4537—HDL-C for TG < 4.5 mmol/L). Non-HDL cholesterol (non-HDL-C) was calculated as follows: non-HDL-C = TC—HDL-C. Glucose was determined by the GOD-PAP method (Roche, Basel, Switzerland) and apoB by immunoturbidimetric method (Tina-quant apoB kits by Roche, Basel, Switzerland). Glycated hemoglobin (HbA1C) levels were measured by ion-exchange chromatography using the ADAMS A1c HA-8180V analyzer (Arkray Corporation, Kyoto, Japan). High-sensitive C-reactive protein (hs-CRP) was assessed by the ultra-sensitive latex immunoturbidimetric method (Tina-quant CRP latex kit by Roche, Basel, Switzerland). Specific antibodies and an immunoradiometric assay in commercially available kits (Immunotech, Marseille, France) were used for insulin and C-peptide concentrations. Tissue plasminogen activator (tPA), plasminogen activator inhibitor-1 (PAI-1) and von Willebrand factor (vWF) were chosen as humoral markers of endothelial damage. VWF antigen was measured by immunoturbidimetric assay (vWF-a, Instrumentation Laboratory, Milan, Italy). Concentrations of t-PA and PAI-1 were determined from human plasma by using ELISA (both by Technoclone, Vienna, Austria).

### 2.4. Statistical Analysis

All values are expressed as means ± standard deviation (SD) or medians and interquartile ranges (Q25–Q75; for data with non-normal distribution). Non-normal distribution was tested by the Shapiro–Wilk test. Differences in variables between the groups were analyzed with the *t*-test and ANOVA for normally distributed variables, the Mann–Whitney U-test and the Kruskal–Wallis test for non-normally distributed variables and the chi-squared (χ^2^) test for categorical variables. Spearman’s coefficient (ρ) was used to express the value of correlation. Multiple regression analysis was used to estimate the relationship between independent and dependent variables. *p* < 0.05 was considered as significant. Statistical analyses were performed using Statistica 12.0 (StatSoft Software Inc., Tulsa, OK, USA). Probability values of *p* < 0.05 were considered as statistically significant.

## 3. Results

### 3.1. Basic Characteristic

A total of 160 patients with T2D participated in this study (109 men, 51 women; age = 58.2 ± 11.7 years). All T2D patients were treated with diet. Of the total number of participants, 72% were on insulin and 91% were on oral antidiabetic drugs (OADs), as follows: metformin 86%, dipeptidyl peptidase-4 inhibitors 36%, gliflozin 14%, sulfonylureas 9% and GLP-1 receptor agonists 7%. Eighty-three percent of subjects were treated with antihypertensive therapy (ACE inhibitors 62%, angiotensin receptor blockers 19%, calcium channel blockers 39%, diuretics 43%, and beta-blockers 35%). Hypolipidemic drugs were administered to 60% of patients (specifically: statins 56%, ezetimibe 16% and fibrates 18%). Among all participants, 35% were smokers.

Table 1 shows the basic characteristics of the participants divided by age into individual quartiles. The oldest patients had, significantly, the highest SBP, prevalence of hypertension, and CVD, but the lowest levels of LDL-C, and hs-CRP.

### 3.2. Vascular Damage Parameters

The results are shown in Table 2 and Figure 1. The oldest patients had, significantly, the highest arterial stiffness parameters, namely AP, AIx-75, Aortic PP and PWV. They also had the highest elevation of vWF and the lowest PAI-1 levels. No significant differences were detected in t-PA.

All markers of arterial stiffness correlated with age (for: AIx-75 ρ = 0.27, AP ρ = 0.43, aortic SP ρ = 0.25, aortic PP ρ = 0.48, and PWV ρ = 0.43). Some parameters correlated with SBP (for: AP ρ = 0.22, aortic SP ρ = 0.75, aortic PP ρ = 0.50, and PWW ρ = 0.27). Aortic SP, in addition, correlated with DBP (ρ = 0.54) and waist circumference (ρ = 0.18); moreover, aortic PP and PWV correlated with insulin levels (ρ = 0.21, ρ = 0.19, respectively). There were no associations between arterial stiffness and parameters of glycometabolic control. PAI-1 correlated positively with BMI (ρ = 0.39), waist circumference (ρ = 0.38), hs-CRP (ρ = 0.23), TG (ρ = 0.40), non-HDL-C (ρ = 0.39), apoB (ρ = 0.35), fasting glucose (ρ = 0.20), C-peptide (ρ = 0.37), and negatively with HDL-C (ρ = −0.27). VWF correlated positively with TG (ρ = 0.26), and negatively with HDL-C (ρ = −0.36). Levels of t-PA correlated only with C-peptide (ρ = 0.20) and PAI-1 (ρ = 0.24). There were no significant correlations between endothelial markers and parameters of arterial stiffness. Multiple regression analyses were performed to identify independent predictors for markers of vascular damage—see Table 3. Age was the only independent predictor for AP. SBP predicted aortic PP, aortic SP (together with DBP), and PWV. Von Willebrand factor was independently predicted by TG and HDL-C; PAI-1 was predicted by C-peptide, and BMI.

## 4. Discussion

Patients with T2D in the highest age quartile have shown the most striking signs of vascular damage. Parameters of arterial stiffness (AP, AIx-75, aortic PP and PWV) were significantly increased in this group compared to younger individuals regardless of glycometabolic control. Arterial stiffness was mainly associated with age and systolic blood pressure. The oldest patients also had the highest levels of vWF, but the lowest levels of PAI-1. Markers of endothelial dysfunction correlated with metabolic parameters, but did not correlate with parameters of arterial stiffness.

Age and blood pressure are among the strongest predictors of arterial stiffness [3,4,7]. Both were significantly and independently associated with PWV in 91% and 90% of conducted studies, respectively, whereas the presence of diabetes was associated with PWV only in 52% of studies. However, even within the studies in which a positive association between diabetes and arterial stiffness was seen, diabetes accounted for only 5% of the variation in PWV [9]. Potential mechanisms of arterial stiffness in diabetes may include the role of advanced glycation end products (AGEs) and nitric oxide (NO) [7]. In the present study, there were no significant differences between age-defined quartiles in glycemic control or duration of diabetes, which primarily determine the development of AGEs. Only blood pressure and age were independently predictors for markers of arterial stiffness. Therefore, we believe that age and systolic blood pressure (still significantly elevated in older patients) are major contributors to arterial stiffness in this cohort of T2D patients. This is consistent with the results of previous studies where diabetes per se (in contrast to hypertension) was only a weak predictor of arterial stiffness [9].

Reduced bioavailability of NO leading to endothelial dysfunction results in impaired vasodilation, increased vascular fibrosis and arterial stiffness [4]. Age-related endothelial dysfunction may affect the arterial network differently depending on the location and type of vessel. Aging results in endothelial dysfunction, particularly in large conduit arteries [16]. It is the involvement of these arteries that most influences the selected parameters of arterial stiffness. In addition to impaired NO-dependent vasodilation, endothelial dysfunction is also manifested by increased production of pro-inflammatory, pro-adhesive and pro-thrombotic molecules, such as vWF, and PAI-1. Although these indicators did not correlate with the examined parameters of arterial stiffness, the oldest patients had the highest levels of vWF, and on the contrary, the lowest levels of PAI-1.

Von Willebrand factor is known to be a more specific marker of endothelial dysfunction than PAI-1 because plasma levels of vWF are exclusively produced by endothelial cells [17,18,19,20], whereas plasma levels of PAI-1 reflect its production not only in the endothelium, but also in adipose tissue and other cells such as megakaryocytes, smooth muscle cells, fibroblasts, monocytes and macrophages [21,22]. Von Willebrand factor plays a key role in platelet adhesion and aggregation, and numerous studies have investigated the relationship between VWF plasma levels and thromboembolic cardiovascular events. VWF typically rises during an acute coronary syndrome, and the extent of this VWF release is an independent predictor of adverse clinical outcomes in these patients. Many lines of evidence suggest that VWF is not only a marker, but also a truly important effector in the pathogenesis of myocardial infarction [18]. A recent meta-analysis showed that plasma vWF levels were significantly higher in T2D patients with CVD than those without CVD [23]. This is consistent with the results of this study; the oldest patients with the highest vWF levels had the highest prevalence of CVD. However, vWF did not correlate with age, but was independently associated with indicators of mixed dyslipidemia (TG and HDL-C levels). Several studies have observed an association between dyslipidemia and VWF in patients with T2D and may point to how dyslipidemia contributes to endothelial damage in this population [24,25].

The lowest PAI-1 levels found in the oldest patients with T2D probably reflect the relatively smaller proportion of adipose tissue in these individuals. Surprisingly, the existence of the lowest PAI-1 levels in the oldest T2D patients of this study contradicts some previous observations [26]. This may be related to the treatment given for diabetes, hypertension, or dyslipidemia, or it might reflect the relatively smaller proportion of adipose tissue in these individuals [27,28]. They also had significantly lower BMIs and hs-CRP levels compared to the youngest patients. PAI-1 correlated with parameters of atherogenic dyslipidemia, abdominal obesity, inflammation, fasting glucose, and C-peptide. C-peptide levels and BMI were independently associated with PAI-1. PAI-1 is produced not only by endothelial cells; a significant amount of PAI-1 is secreted by adipose tissue [22,28,29]. Significant correlations of PAI-1 with different indicators of visceral obesity (e.g., BMI, waist circumference, waist-to-hip ratio, etc.), markers of insulin resistance, and adverse metabolic profile have been reported [30,31]. Elevated plasma PAI-1 levels in obese subjects can be normalized by weight-loss diet or bariatric surgery [32,33,34]. Adipose tissue is responsible for the secretion of various pro-inflammatory cytokines, adipokines and markers of chronic inflammation, which are associated with the development of insulin resistance. Thus, the association of PAI-1 with obesity and diabetes may reflect a confounding association of one or more other inflammatory markers. However, growing evidence supports a potential association between PAI-1 and the development of T2D, regardless of other established risk factors for diabetes [35]. This can apply especially to younger individuals who are more obese and have more pronounced chronic low-grade inflammation.

A limitation of this study is its cross-sectional and non-randomized design, especially in relation to drugs that potentially affect both parameters of arterial stiffness and indicators of endothelial dysfunction. Only 14% of study participants were treated with gliflozins and 7% with GLP-1 receptor agonists, which did not allow statistical evaluation with significant power. Conversely, a relatively high number of participants were treated with drugs (ACE inhibitors, sartans, statins, or their combination), potentially affecting arterial stiffness and/or endothelial function. Moreover, the effect of drugs on vascular damage is likely to be time- and dose-dependent. Thus, larger prospective studies are needed to determine whether better glycometabolic control can improve arterial elasticity in elderly patients with T2D and to find the possible role of different classes of other antidiabetic drugs.

## 5. Conclusions

Age, together with systolic blood pressure, seems to be the main determinant of arterial stiffness in patients with T2D, regardless of glycometabolic control. This is consistent with the results of previous studies where diabetes per se was only a weak predictor of arterial stiffness. An unfavorable metabolic profile is mainly related to endothelial dysfunction. The lack of correlation between markers of endothelial dysfunction and arterial stiffness suggests a differential contribution of cardiometabolic risk factors to various markers of vascular damage in T2D patients throughout their lifetime. This may require a different, age-specific therapeutic approach.

## Figures and Tables

**Figure 1 metabolites-13-00382-f001:**
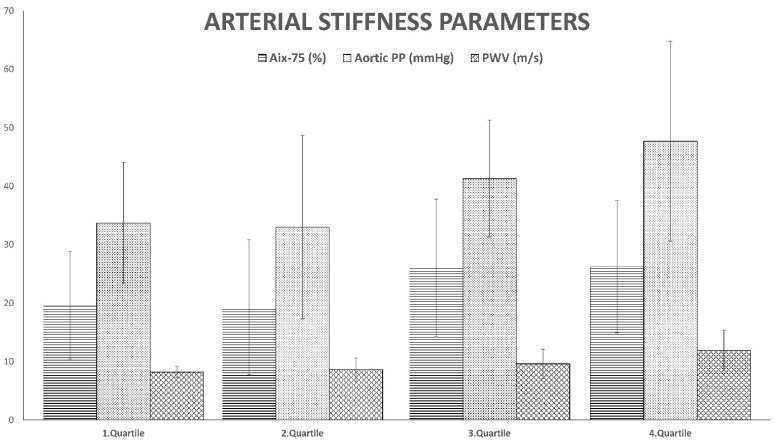
Selected parameters of arterial stiffness in individual quartiles according to age. AIx-75 = augmentation index normalized for a heart rate of 75 bpm; aortic PP = aortic pulse pressure; PWV = pulse wave velocity.

**Table 1 metabolites-13-00382-t001:** Basic clinical and laboratory characteristics in individual quartiles according to age.

	1. Quartile (*n* = 39)	2. Quartile (*n* = 38)	3. Quartile (*n* = 41)	4. Quartile (*n* = 42)	Corrected *p*-Value
age (years)	42.1 ± 4.5	51.6 ± 1.4	59.2 ± 3.0	69.8 ± 3.8	-
female (percentage)	15 (38%)	11 (29%)	12 (29%)	13 (31%)	n.s.
BMI (kg/m^2^)	34.1 ± 5.3	31.8 ± 5.3	32.7 ± 5.2	30.9 ± 4.0	n.s.
waist (cm)	115.3 ± 17.6	110.4 ± 14.3	113.5 ± 14.2	110.0 ± 11.7	n.s.
SBP (mmHg)	130.2 ± 17.1 ^d^	124.4 ± 14.5 ^d^	132.0 ± 14.0 ^d^	140.3 ± 15.5 ^a,b,c^	0.002
DBP (mmHg)	79.3 ± 9.1	83.1 ± 10.3	79.6 ± 9.1	81.1 ± 10.7	n.s.
TC (mmol/L)	5.0 ± 1.9	4.8 ± 1.7	4.5 ± 1.0	4.2 ± 1.3	n.s.
LDL-C (mmol/L)	2.6 ± 0.7 ^d^	2.5 ± 0.8	2.5 ± 0.7	2.0 ± 1.1 ^a^	0.020
HDL-C (mmol/L)	1.0 ± 0.3	1.0 ± 0.3	1.1 ± 0.3	1.1 ± 0.4	n.s.
TG (mmol/L)	2.1 (1.6–3.0)	1.8(1.3–2.6)	1.7 (1.2–2.5)	2.0 (1.3–2.9)	n.s.
non-HDL-C (mmol/L)	3.9 ± 1.9	3.8 ± 1.8	3.5 ± 1.0	3.1 ± 1.4	n.s.
apoB (g/L)	1.1 (0.9–1.3)	1.0 (0.9–1.2)	1.0 (0.8–1.3)	0.9 (0.7–1.0)	n.s.
FPG (mmol/L)	7.8 (6.2–9.9)	8.3 (6.9–11.0)	7.4 (6.1–11.2)	8.1 (7.7–11.0)	n.s.
HbA_1c_ (mmol/mol)	61.5 (43.5–73.3)	71.5 (49.8–97.5)	64.0 (44.9–95.5)	69.3 (49.5–82.9)	n.s.
Insulin (mIU/L)	19.2 (13.0–44.8)	20.1(12.3–36.1)	17.5 (11.5–36.8)	29.7 (13.6–73.6)	n.s.
C-peptide (pmol/L)	846 (565–1199)	800 (607–1123)	940 (617–1275)	746 (423–1083)	n.s.
hs-CRP (mg/L)	4.1 (2.2–8.4) ^d^	4.1 (2.1–8.4) ^d^	4.4 (1.8–8.8) ^d^	1.6 (0.8–3.5) ^a,b,c^	0.007
dd (years)	6.4 ± 3.5	6.0 ± 3.1	10.0 ± 6.5	9.1 ± 6.1	n.s.
CVD (percentage)	1 (3%)	9 (23%)	12 (29%)	21 (50%)	0.001
hypertension (percentage)	29 (74%)	35 (92%)	37 (90%)	40 (95%)	0.002

BMI = body mass index; SBP = systolic blood pressure; DBP = diastolic blood pressure; TC = total cholesterol; LDL-C = LDL cholesterol; HDL-C = HDL cholesterol; TG = triglycerides; non-HDL-C = non-HDL cholesterol; apoB = apolipoprotein B; FPG = fasting plasma glucose; HbA_1c_ = glycated hemoglobin A_1c_; hs-CRP = high sensitive C-reactive protein; dd = diabetes duration; CVD = cardiovascular disease (coronary heart disease, myocardial infarction, stroke, peripheral artery disease, revascularization) in personal history. Values are expressed as means ± standard deviation (SD) or medians and interquartile ranges (Q25–Q75; for data with non-normal distribution) or as the percentage share for categorical variables. Difference between groups was tested by one-way ANOVA for normally distributed variables or Kruskal–Wallis test otherwise. Holm–Bonferroni procedure was used to correct for multiple testing. In cases of significant difference, pairwise tests were performed by *t*-test for normally distributed variables or the Mann–Whitney U-test otherwise. Chi-square test was used for categorial data. Significant differences (*p* < 0.05) between group: **^a^** = vs. 1. Quartile; **^b^** = vs. 2. Quartile; **^c^** = vs. 3. Quartile; **^d^** = vs. 4. Quartile; n.s. = non-significant.

**Table 2 metabolites-13-00382-t002:** Vascular damage parameters in individual quartiles according to age.

	1. Quartile (*n* = 39)	2. Quartile (*n* = 38)	3. Quartile (*n* = 42)	4. Quartile (*n* = 41)	Corrected *p*-Value
v-WFa (%)	121.7 ± 42.5 ^b,c,d^	134.8 ± 50.0 ^a,d^	129.0 ± 32.9 ^a,d^	153.7 ± 51.9 ^a,b,c^	0.010
PAI-1 (ng/mL)	90.0 ± 44.9	100.9 ± 48.3 ^d^	88.4 ± 43.9	81.8 ± 47.5 ^b^	0.048
t-PA (ng/mL)	2.8(2.6–3.1)	2.7(2.3–3.8)	3.0(2.5–3.6)	2.8(2.4–3.7)	n.s.
AP (mmHg)	6.7 ± 4.4 ^c,d^	7.0 ± 5.7 ^c,d^	11.2 ± 6.6 ^a,b^	13.4 ± 8.6 ^a,b^	<0.001
AIx-75 (%)	19.6 ± 9.2 ^c,d^	19.3 ± 11.5 ^c,d^	26.0 ± 11.8 ^a,b^	26.2 ± 11.3 ^a,b^	0.005
aortic SP (mmHg)	118.0 ± 15.4 ^b^	108.7 ± 17.2 ^a,c,d^	123.8 ± 15.3 ^b^	122.6 ± 18.1 ^b^	0.001
aortic PP (mmHg)	33.7 ± 10.4 ^c,d^	33.0 ± 15.7 ^c,d^	41.3 ± 10.0 ^a,b,d^	47.7 ± 17.1 ^a,b,c^	<0.001
PWV (m/s)	8.2(7.7–9.8) ^c,d^	8.6(7.0–10.6) ^d^	9.6(7.9–11.7) ^d^	11.9(10.1–14.3) ^a,b,c^	<0.001

PAI-1 = plasminogen activator inhibitor-1; t-PA = tissue plasminogen activator; vWF = von Willebrand factor; AIx-75 = augmentation index normalized for a heart rate of 75 bpm; AP = augmentation pressure; aortic SP = aortic systolic pressure; aortic PP = aortic pulse pressure; PWV = pulse wave velocity. Values are expressed as means ± standard deviation (SD) or medians and interquartile ranges (Q25–Q75; for data with non-normal distribution) or as the percentage share for categorical variables. Difference between groups was tested by one-way ANOVA for normally distributed variables or Kruskal–Wallis test otherwise. Holm–Bonferroni procedure was used to correct for multiple testing. In cases of significant difference, pairwise tests were performed by *t*-test for normally distributed variables or the Mann–Whitney U-test otherwise. Significant differences (*p* < 0.05) between group: **^a^** = vs. 1. Quartile; **^b^** = vs. 2. Quartile; **^c^** = vs. 3. Quartile; **^d^** = vs. 4. Quartile; n.s. = non-significant.

**Table 3 metabolites-13-00382-t003:** Multiple regression analyses of independent factors affecting selected vascular damage markers as dependent variables.

	Pulse Wave Velocity					
	* **Unstandardized coefficients** *		* **Standardized coefficients** *		* **t** *	* **Sig.** *
	* **B** *	* **SE** *	* **Beta** *	* **SE** *		
age	0.2075	0.2473	0.0480	0.0573	0.8391	0.4027
SBP	0.6052	0.2484	0.0603	0.0247	2.4365	0.0159
insulin	0.0046	0.0604	0.0010	0.0132	0.0755	0.9399
	**Aortic pulse pressure**					
	*Unstandardized coefficients*		*Standardized coefficients*		*t*	*Sig.*
	*B*	*SE*	*Beta*	*SE*		
age	0.0651	0.0482	0.1366	0.1010	1.3524	0.1782
SBP	0.9357	0.0484	0.8439	0.0436	19.3454	0.0000
insulin	−0.0096	0.0118	−0.0191	0.0233	−0.8194	0.4138
	**Aortic systolic pressure**					
	*Unstandardized coefficients*		*Standardized coefficients*		*t*	*Sig.*
	*B*	*SE*	*Beta*	*SE*		
age	0.0804	0.0486	0.1686	0.1020	1.6538	0.1002
waist	0.0664	0.0615	0.0703	0.0651	1.0799	0.2818
SBP	0.6147	0.1081	0.5544	0.0975	5.6880	0.0000
DBP	0.2354	0.0986	0.3469	0.1452	2.3883	0.0181
	**Augmentation index-75**					
	*Unstandardized coefficients*		*Standardized coefficients*		*t*	*Sig.*
	*B*	*SE*	*Beta*	*SE*		
age	0.3732	0.0756	0.2501	0.0507	4.9361	0.0000
SBP	0.1092	0.0756	0.0481	0.0333	1.4442	0.1507
	**Von Willebrand factor**					
	*Unstandardized coefficients*		*Standardized coefficients*		*t*	*Sig.*
	*B*	*SE*	*Beta*	*SE*		
TG	0.2278	0.0383	7.4126	1.2450	5.9537	0.0000
HDL-C	0.7713	0.0383	101.644	5.0416	20.1609	0.0000
	**Plasminogen activator inhibitor-1**					
	*Unstandardized coefficients*		*Standardized coefficients*		*t*	*Sig.*
	*B*	*SE*	*Beta*	*SE*		
BMI	0.5258	0.1442	1.5399	0.4222	3.6474	0.0004
waist	−0.0881	0.4868	−0.0746	0.4122	−0.1809	0.8567
hs-CRP	0.0099	0.0503	0.1427	0.7236	0.1972	0.8440
TG	0.0923	0.0700	1.9695	1.4931	1.3190	0.1891
HDL-C	−0.1693	0.1014	−14.6352	8.7653	−1.6697	0.0970
non-HDL-C	0.0584	0.1791	1.4434	4.4267	0.3261	0.7448
apo-B	0.1866	0.2026	17.1592	18.6299	0.9211	0.3585
FPG	0.1331	0.0941	1.3535	0.9562	1.4155	0.1590
C-peptide	0.2081	0.0697	0.0187	0.0063	2.9859	0.0033

SBP = systolic blood pressure; DBP = diastolic blood pressure; TG = triglycerides; HDL-C = HDL cholesterol; non-HDL-C = non-HDL cholesterol; apoB = apolipoprotein B; FPG = fasting glucose.

## Data Availability

The data used in this manuscript are available upon request.

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
