# Peer review of "Vascular Damage and Glycometabolic Control in Older Patients with Type 2 Diabetes"

_metabolites, 2023, doi:10.3390/metabo13030382_

Round 1
Reviewer 1 Report
The authors performed an excellent cross-sectional study which successfully deduced that age and systolic blood pressure are the major determinants of arterial stiffness in Type 2 Diabetes patients. They also show that this correlation is independent of glycometabolic control in these patients. They also show that endothelial dysfunction is primarily linked to metabolic parameters. In this study coincidentally the oldest patients had highest vWF levels and they also had most incidence of cardiovascular disease. vWF independently associated with markers like TG and HDL-C levels.
Overall, despite the limitations like not factoring in the drugs effecting parameters of arterial stiffness and signs of endothelial dysfunction. This study provides an important direction of looking at systolic blood pressure for arterial stiffness in aged patients.
minor error :
line 258 : "......vWF did nor correlated......." correct it to "vWF did not correlate with age"
Author Response
Thank you very much for your valuable comments.
Comment #1 minor error: line 258 : "......vWF did nor correlated......." correct it to "vWF did not correlate with age"
Response: This error was corrected.
Reviewer 2 Report
Dear Authors,
Thank you for the opportunity to review the article entitled „Vascular damage and glycometabolic control in older patients 2 with type 2 diabetes” which addresses the description of the relationship between parameters of arterial stiffness, markers of endothelial damage and parameters of glucometabolic control.
WHAT IS THE MAIN QUESTION ADDRESSED BY THE RESEARCH?
In the lines 50-53 is described the aim of the study as ” to compare markers of glycometabolic control and vascular damage in patients with type 2 diabetes (T2D) according to age and to determine the relationship between parameters of arterial stiffness together with markers of endothelial damage and parameters of glycometabolic control and other risk factors for CVD”.
Please formulate the aim and hypothesis of the study in a clearer way. E.g. whether the parameteers of arterial stiffness are influenced of the metabolic control in patients with diabetes.
IS THE MANUSCRIPT SCIENTIFICALLY SOUND AND IS THE EXPERIMENTAL DESIGN APPROPRIATE TO TEST THE HYPOTHESIS?
The study is correctly designed and technically sound. However, as diabetes is a well known major cardiovascular risk factor, it would have been more appropriate to include a control group of people without diabetes with similar age and gender structure, in this study. Were patients with cardiovascular disease included in this study? Was the duration of diabetes considered as a potential cofounder of the analisys included?
With the increase in the number of cardiovascular risk factors, the risk of death or non-fatal cardiovascular event increases. It would have been useful to stratify the analysis in relation to the the categories of cardiovascular risk in patients with diabetes (ESC Guidelines, European Journal of Preventive Cardiology (2021), as a confounding factor of the relationship between parameteers of arterial stiffness are influenced of the markers of metabolic control.
IS THE TOPIC ORIGINAL OR RELEVANT IN THE FIELD? The subject under study is certainly important as the incidence of diabetes is raising and cardiovascular disease is the main cause of death in the diabetic patients. The article presents interesting results but, it must be improved, especially for the small sample size, and some methodological concerns.
In this study four groups formed base on age of the participants were compared. Can you include the results of the power analyses in the article?
The methods used in this research are well described and provide sufficient details to be understand.
The disscutions are relevant the paper's focus area and address the findings of the research in relation with other studies and also propose potential explanation for the relation between arterial stiffness and markers of glycometabolic control in patients with type 2 diabetes.
Thank you for your esteemed efforts in increasing our collective knowledge about potential mecanism of the strong association between diabetes and cardiovascular disease.
Author Response
Thank you very much for your valuable comments.
Comment #1 In the lines 50-53 is described the aim of the study as ” to compare markers of glycometabolic control and vascular damage in patients with type 2 diabetes (T2D) according to age and to determine the relationship between parameters of arterial stiffness together with markers of endothelial damage and parameters of glycometabolic control and other risk factors for CVD”. Please formulate the aim and hypothesis of the study in a clearer way.
Response: The study aim was formulated more simply and clearly.
Comment #2 However, as diabetes is a well-known major cardiovascular risk factor, it would have been more appropriate to include a control group of people without diabetes with similar age and gender structure, in this study. Were patients with cardiovascular disease included in this study? Was the duration of diabetes considered as a potential cofounder of the analisys included? With the increase in the number of cardiovascular risk factors, the risk of death or non-fatal cardiovascular event increases. It would have been useful to stratify the analysis in relation to the the categories of cardiovascular risk in patients with diabetes (ESC Guidelines, European Journal of Preventive Cardiology (2021), as a confounding factor of the relationship between parameteers of arterial stiffness are influenced of the markers of metabolic control.
Response: You're right. However, this study was designed to be cross-sectional (not case-control) to compare T2D patients by age group. This study included patients with cardiovascular disease - see Table 1. There were no differences between groups in duration of diabetes (see Table 1) and it did not correlate with estimated markers of vascular damage, therefore this factor was not included in the multiple regression analysis. We stratified participants by age, further stratification (based on cardiovascular risk or medication only) would have been accompanied by fragmentation of the comparison group and would have reduced statistical significance.
Comment #3 In this study four groups formed base on age of the participants were compared. Can you include the results of the power analyses in the article?
Response: Sample size determination was performed to obtain adequate power to detect statistical significance. Sample size estimation (N) was based on means between groups (Null hypothesis Ho: m1 = m2 vs. alternative hypothesis Ha: m1 = m2 + d) and power analysis. It was done using this formula N = (r+1)*(Za+Zp)2*σ2 / r*d2 [1]. For the same sample size, r = 1, so N = 2*(Zα+Zβ)2*(σ/d)2. σ represents the standard deviation and d is the difference of group means. A statistical power value of at least 80% was desired, so a β level of 0.20 and a corresponding Zβ = 0.84 were used for type II error. For Type I error, an α level of 0.05 and a corresponding Zα = 1.96 were used. Therefore N = 15.68*(σ/d)2 for these conditions. The sample size of participants in the investigated groups was about 40. The ratio σ/d must be lower than 1.60 in this case. This condition was met for most of the measured variables. Thus, the number of participants in each group was large enough to avoid a type II error.
- Suresh K, Chandrashekara S. Sample size estimation and power analysis for clinical research studies. J Hum Reprod Sci 2012; 5:7-13.
Reviewer 3 Report
The article reports about a study performed on age related subgroups diabetic paients.Authors found that stifness of vessels correlated with age and blood pressure especially in the older age patient`s group.Interestingly the junger patients showed higher CRP-serum level.Vascular damage parameters were hgher in the older age group.
Comment:
1.there is no control groups
2.D-dimer measurement would have been more appropriate as a possible parameter for endothelial damage.Angiotensinogen (also synthesized in the liver by hepatocytes) serum level would also be interesting
3.vWF is synthesisized exclusively and not "mostly" by endothelial cells.Serum level is most probably due to the synthesis af this protein in the largest organ,namely the liver(Knittel T 1995,Baruch Y 2002 J Hepatol,Barch Y 2004
Author Response
Thank you very much for your valuable comments.
Comment #1.there is no control groups
Response: You are correct, healthy controls were not included. However, this study was designed to be cross-sectional (not case-control) to compare T2D patients by age group. Only patients with T2D were included.
Comment # 2.D-dimer measurement would have been more appropriate as a possible parameter for endothelial damage.Angiotensinogen (also synthesized in the liver by hepatocytes) serum level would also be interesting
Response: You are right, there are a number of other parameters to assess vascular damage. But this study was only focused on arterial stiffness indicators and endothelial hemostatic markers, so we only have data for t-PA, PAI-1 and vWF.
Comment #3.vWF is synthesisized exclusively and not "mostly" by endothelial cells.Serum level is most probably due to the synthesis af this protein in the largest organ,namely the liver(Knittel T 1995,Baruch Y 2002 J Hepatol,Barch Y 2004
Response: The text of the manuscript has been corrected and references added.
Reviewer 4 Report
This paper deals with a topic of great importance and clinical evidence, it is well written and has an updated reference section.
Unfortunately it merely represents confirmatory data about what is well known from decades: "Age together with systolic blood pressure seems to be the main determinant of arterial stiffness..."
What is more problematic is the statistical approach here employed. The linear relationship is highly wrong in this contest, since the authors hide in this way all the possible confounders.
A correct statistical approach would have been represented by a multiple regression analysis, corrected for confounders to analyze the actual relationships (while the linear regression cannot).
I am afraid to say that the paper is to be rejected.
Author Response
Thank you very much for your valuable comments.
Comment #1 Unfortunately it merely represents confirmatory data about what is well known from decades: "Age together with systolic blood pressure seems to be the main determinant of arterial stiffness..."
Response: You are right, age and systolic blood pressure are undoubtedly the strongest predictors of arterial stiffness. However, we could assume that glycometabolic control will also play an important role in individuals with diabetes. The results of the study did not show its significant effect on the indicators of the vascular wall (in contrast to the parameters of endothelial dysfunction). We have been invited to present a study focusing on vascular damage and glycometabolic control in elderly patients with diabetes, and we still believe that despite the growing evidence of data in this area, the results may be of interest to the readers of "Metabolites".
Comment #2 The linear relationship is highly wrong in this contest, since the authors hide in this way all the possible confounders. A correct statistical approach would have been represented by a multiple regression analysis, corrected for confounders to analyze the actual relationships (while the linear regression cannot).
Response: Multiple regression analysis is necessary to correct confounders to analyze the actual independent relationships. Multiple linear regression is used to estimate the relationship between two or more independent variables and one dependent variable [1]. And that's exactly what we did. First, a correlation analysis (expressed by Spearman correlation coefficients) was made and then the dependent variables were tested on the possible independent variables (which were correlated in the univariate analysis) using multiple regression analysis - see Table 1. You are right that that term “multiple” is better than “multivariate” for the situation, where the number of predictors were entered the model for a single outcome [2]. The manuscript text was corrected.
References:
- https://www.scribbr.com/statistics/multiple-linear-regression/
- https://stats.stackexchange.com/questions/2358/explain-the-difference-between-multiple-regression-and-multivariate-regression
Round 2
Reviewer 3 Report
The author write that the aim of the study was to investigate the endothelial damage (lines 50-51).they should therefore add to the text the answers they gave to my criticisms and also add that the measurement of D-Dimer would have been more appropriate for this purpose.
Authors should also still have plasma in the refrigerator and measure that marker.This would increase the quality of the manuscript.
In addition this point should be added to the discussion.
Author Response
Thank you very much for your valuable comments.
Comment #1. The author write that the aim of the study was to investigate the endothelial damage (lines 50-51).they should therefore add to the text the answers they gave to my criticisms and also add that the measurement of D-Dimer would have been more appropriate for this purpose.
Authors should also still have plasma in the refrigerator and measure that marker.This would increase the quality of the manuscript.
In addition this point should be added to the discussion.
Response: We are very sorry, but all plasma aliquots were destroyed after measurement, so we are unable to assess D-dimer levels. You are right, D-dimer is probably one of the independent predictors of CVD risk. Now this fact is mentioned in the discussion and corresponding references have been added. We will include this parameter in our future research. Thank you for your recommendation.
Reviewer 4 Report
Unfortunately, in my opinion, the serious flaws of the paper remain even after revision. This is why my suggestion is to shorten the paper and to send elsewhere as a short communication or letter.
Author Response
Comment #1 Unfortunately, in my opinion, the serious flaws of the paper remain even after revision. This is why my suggestion is to shorten the paper and to send elsewhere as a short communication or letter.
Response: Thank you for your opinion. We have tried to respond to your previous comments and correct the manuscript. Now, unlike other reviewers - they want to expand the discussion, you recommend shortening the text and submitting it elsewhere. We were invited to submit this manuscript based on the study abstract, and we still believe that the results may be of interest to Metabolites readers. Therefore, we will leave this decision to the editors of the journal.
Round 3
Reviewer 3 Report
Authors have satisfactorly addressed my concerns
Author Response
Thank you very much for your review.